



# The CISE-LOCEAN sea water isotopic database (1998-2021)

Gilles Reverdin[1], Claire Waelbroeck[1], Catherine Pierre[1], Camille Akhoudas[1], Giovanni Aloisi[2], Marion Benetti[1], Bernard Bourlès[3], Magnus Danielsen[4], Jérôme Demange[1], Denis Diverrès[3], Jean-Claude Gascard[1], Marie-Noëlle Houssais[1], Hervé Le Goff[1], Pascale Lherminier[5], Claire Lo Monaco[1], Herlé Mercier[5], Nicolas Metzl[1], Simon Morisset[6], Aïcha Naamar[1], Thierry Reynaud[5], Jean-Baptiste Sallée[1], Virginie Thierry[5], Susan E. Hartman[7], Edward W. Mawji[7], Solveig Olafsdottir[4], Torsten Kanzow[8], Antje Voelker[9,10], Igor Yashayaev[11]

Sorbonne University, LOCEAN - IPSL, CNRS–IRD–MNHN, Paris, France
Université de Paris, Institut de Physique du Globe de Paris, CNRS, 75005 Paris, France
UMS IMAGO, IRD, Plouzané, France
Marine and Fresh Water Institute, Iceland
University of Brest, LOPS, IUEM, UBO–CNRS–IRD–Ifremer, Plouzané, France
Amundsen Science, Québec, Canada
National Oceanogaphy Center, Southampton, UK
MARUM/Alfred Wegener Institute for Polar and Marine Research, Bremerhaven, Germany
Instituto Português do Mar e da Atmosfera, Lisbon, Portugal
Centro de Ciencias do Mar, Faro, Portugal
Bedford Institute of Oceanogaphy, Dartmouth, Nova Scotia Canada

Corresponding author: Gilles Reverdin, gilles.reverdin@locean.ipsl.fr





Abstract
The characteristics of the CISE-LOCEAN sea water isotope data set ($\delta^{18}O$, $\delta^2H$, later designed
as $\delta D$) are presented. This data set covers the time period from 1998 to 2021 and currently
includes close to 8000 data entries, all with $\delta^{18}O$, three quarters of them also with $\delta D$, associated
with a time and space stamp and usually a salinity measurement. Until 2010, samples were
analysed by isotopic ratio mass spectrometry, and since then mostly by cavity ring-down
spectroscopy (CRDS). Instrumental uncertainty on individual data in this dataset is usually with
a standard deviation as low as 0.03 / 0.15‰ for $\delta^{18}O$ and $\delta D$. An additional uncertainty is related
to uncertain isotopic composition of the in-house standards that are used to convert daily data
into the VSMOW scale. Different comparisons suggest that since 2010 the latter have remained
within at most 0.03/0.20‰ for $\delta^{18}O$ and $\delta D$.  Therefore, combining the two suggests a
standard deviation of at most 0.05 / 0.25‰ for $\delta^{18}O$ / $\delta D$.
Finally, for some samples, we find that there has been evaporation during collection and
storage, requiring adjustment of the isotopic data produced by CRDS, based on d-excess. This
adds an uncertainty on the adjusted data of roughly 0.05 / 0.10‰ on $\delta^{18}O$ and $\delta D$. This issue of
conservation of samples is certainly a strong source of quality loss for parts of the database, and
'small' effects may have remained undetected.
The internal consistency of the database can be tested for subsets of the dataset, when time
series can be obtained (such as in the southern Indian Ocean or North Atlantic subpolar gyre).
These comparisons suggest that the overall uncertainty of the spatially (for a cruise) or
temporally (over a year) averaged data is on the order of or less than 0.03 / 0.15‰ for $\delta^{18}O$ /
$\delta D$. On the other hand, 17 comparisons with duplicate sea water data analysed in other
laboratories or with other data sets in deep regions suggest a larger scatter. When averaging the
17 comparisons done for $\delta^{18}O$, we find a difference close to the adjustment applied at LOCEAN
to convert salty water data from the activity to the concentration scale. Such a difference is
expected, but the scatter found suggests that care is needed when merging datasets from
different laboratories. Examples of time series in the surface North Atlantic subpolar gyre
illustrate the temporal changes in water isotope composition that can be detected with a
carefully validated dataset.



1.  Introduction
Stable isotope analyses of ocean water ($\delta^{18}O$, $\delta^2H$ later designed as $\delta D$) were first discussed by
Craig and Gordon (1965) as tracers of water masses, and of the different components of the
global hydrological cycle, in particular the signals gained through evaporation, precipitation,
the interaction with sea ice, and continental water inputs, for example from the ice caps of
Greenland and Antarctica, and ice shelves. Sea water stable isotopes have been used to verify
ocean model circulation and characterize processes controlling their spatial variability (Xu et
al., 2012). Sea water isotopes have also been used to provide information on what controls the
oxygen isotopic ratio of calcite plankton shells, in order to reconstruct past ocean salinity and
circulation. The GEOSECS program (Östlund et al., 1987) provided the first consistent global
dataset of sea water isotopes, but with a limited data coverage. The Global Seawater Oxygen-
18 Database at GISS (Schmidt et al, 1999) has assembled most water isotope data collected
prior to 1998, with an effort to homogenize the dataset, when possible, by estimating biases
based on multiple measurements of deep-water samples (Schmidt, 1999; Bigg and Rohling,
1999). A large part of the early analyses was done by isotope ratio mass spectrometry (IRMS)
and more recently using cavity ring-down spectrometry (CRDS). Walker et al. (2016)
illustrated that the two measurement techniques can provide equivalent results with no obvious
biases.
Since 1998, the CISE-LOCEAN isotopic platform facility has measured sea water isotopic
composition of samples collected on a series of oceanographic cruises or ships of opportunity,
mostly in the North Atlantic, in the equatorial Atlantic, in the southern Indian Ocean and
Southern Ocean. This LOCEAN data set of the oxygen and hydrogen isotopes ($\delta^{18}O$ and $\delta D$)
of marine water covers the period 1998 to 2021, and is ongoing. Most data prior to 2010 (only
$\delta^{18}O$) were produced using an Isoprime IRMS coupled with a Multiprep system (dual inlet
method), whereas most data since 2010 (and a few earlier data) were obtained by CRDS, usually
with a Picarro L2130-i, or less commonly on a Picarro L2120-i. Occasionally, some data were
also run on an Isoprime IRMS coupled to a GasBench (dual inlet method) at the university of
Iceland (Reykjavik). There are also a few pairs of samples measured on both systems. Most of
these LOCEAN data are not currently included in the Global Seawater Oxygen-18 Database at
GISS (Schmidt, Bigg and Roehling, 1999), except for the 1998 OISO cruise data (NB: earlier
datasets measured by co-author C. Pierre on other mass spectrometers preceding the current
IRMS are included in the GISS database). Subsets of the LOCEAN data have been used in
publications (Akhoudas et al., 2020, 2021; Benetti et al., 2016; Benetti et al., 2017; Benetti et
al., 2015), where the subsets correspond to measurements at LOCEAN over a short period with
specific instrumental and analysis protocols. A regional surface Atlantic subset of the data was
also presented in Reverdin et al. (2018).
Here, we review the errors and uncertainties in this published dataset (Waterisotopes-CISE-
LOCEAN, 2021), and the extent the overall dataset of $\delta^{18}O$, $\delta D$, and d-excess (d-excess=$\delta D$ - 8
$\delta^{18}O$) presented as per mil V-SMOW, is internally consistent. We will also discuss how the
CISE-LOCEAN sea water isotopic database compares with other datasets, in particular GISS,
and provide some overall statistics on the number of data and their distribution.
2.  Uncertainties
We will first review the different sources of uncertainties relevant for this dataset, before
discussing the scale used and correction and flagging of data.





Uncertainties in the data reported originate from:
- the water collection and storage in bottles (Sect. 2.1)
- the uncertainties resulting from the experimental laboratory set-up and analysis protocols
(Sect. 2.2)
- the uncertainties on the internal standards which are used in the experimental set-up (Sect.
86   2.3)
2.1 Collection and storage
At LOCEAN, we have mostly used glass-tinted bottles (volume 20 or 30 ml) with a hard cap
including an internal rim to minimize water exchange through the cap (referred to later on as
'common' cap). No independent internal stopper or insert is used. For some, but not all, cruises,
the cap has been secured with parafilm after sample collection. When arriving in the laboratory,
samples are commonly stored in a cold room or in a refrigerator at 4°C, except when the
analysis is expected within 3 months after the arrival of the samples. The analysis has
commonly been done within 1 year – 18 months after collection, and for some subsets such as
for SURATLANT (Reverdin et al., 2018b), the analysis was usually done within 3 months after
collection. However, due to various changes at LOCEAN, there has been at times a long
backlog, with some samples having been stored in the cold room for 5 years or more. The
longest storage time was for OISO-18 data collected in 2010 and analyzed 9 years later in 2019.
Storage time was also very long for most samples of cruises OISO-21, OISO-22, OISO-23,
OISO-25 and OISO-26 (South Indian Ocean, 2012 to 2016).
We tested whether the samples in 'common' cap bottles change during storage by aging three
reference waters of the same deep equatorial Atlantic origin over two years in a laboratory room
which is not air-conditioned and without securing the 'common' caps with parafilm. Water is
extracted every three months for isotopic analysis, which so far over 23 months has not revealed
any significant drift, certainly not larger than 0.02/0.1 ‰ in $\delta^{18}O$ / $\delta D$. We expect that drifts
would be even smaller when samples are stored at 4°C or with parafilm, if the caps are properly
tightened.
In 2019, new caps were introduced which were not rigid and would often not provide a tight
seal, with very large sample evolution over less than a year, sometimes reaching close to 1 ‰
in $\delta^{18}O$. This was the case in particular for the samples collected on M/V Nuka Arctica in April
2019 resulting in 32% of samples with suspected breathing (indicated by unexpected low d-
excess and high $\delta^{18}O$; we verified this hypothesis by aging water in bottles with this cap, which
also showed large drifts after three months at room temperature).
Even for bottles with the 'common' caps, issues of poor conservation have been suspected in
some cases, in particular after long storage (typically, for 5 years or more). There is also the
possibility that breathing has happened during transport, in particular when the samples have
experienced very high temperatures, for instance for cruises ending in tropical ports and with
long-time storage in containers. This was probably the case for samples from the EUREC4A-
OA cruise collected in February 2020 (Stevens et al, 2021) with an almost two-months storage
in a container placed without sun-shielding in Pointe-à-Pitre (Guadeloupe, France), for which
close to 22% of the bottles with no parafilm securing the cap are suspected to have breathed
(during analysis, we noticed that the cap was often not tightly closed; their isotopic values also
contrasted with the ones from special tightly closed nutrient vials pasteurized at 80°C for 40
minutes after collection that did not present any anomalous d-excess). There are also other
subsets with data presenting obvious breathing. The extreme case is for samples collected on
M/V Nuka Arctica in 2018-2019, for which we suspect evaporation for 20% of the water



samples. In this case, the water was transferred from salinity bottles during the salinity analysis
to be stored in bottles with the 'common' cap, where they stayed for close to 18 months before
been analyzed.

2.2 Laboratory measurements

2.2.1 Method and protocol of analysis
Until 2011, the seawater samples $\delta^{18}O$ was directly measured on an Isoprime IRMS coupled to
a Multiprep system (dual inlet method). A typical run lasted more than 24 hours, with a few in-
house/internal standards interspersed in the run. Drifts in the values corresponding to the
internal standard used at the time ('Eau de Paris', referred to as EDP) were corrected for,
assuming that the correction is not dependent on salinity or isotopic value. When checking the
records, we found that $\delta^{18}O$ drift between successive EDP samples were often larger than 0.05
‰. Uncertainty on correcting these drifts probably is on the order of 0.05 ‰.

Since 2011, CRDS has been used, which simultaneously measures the samples $\delta^{18}O$ and $\delta D$.
Each sample is vaporized, then injected in the cavity, a process repeated 6 to 12 times. The
average and SD of the sample $\delta^{18}O$ and $\delta D$ are computed out of the last (2 to 8) injections after
stabilization is reached (Skzypek and Ford, 2014). This way, memory effects due to
contamination by the previous sample are small, in particular for $\delta^{18}O$ (Lis et al., 2008;
Skrzypek and Ford, 2014; Vallet-Coulomb et al., 2021). The SD computed on the 2 to 8 selected
injections is taken as an estimate of the instrumental error on the sample $\delta^{18}O$ and $\delta D$
measurements.

When a Picarro CRDS was first used at LOCEAN in 2011-2015, samples were distilled, and
the measurement was thus done on freshwater (see Benetti et al., 2017a, for the average effect
of the distillation on isotopic composition). Since 2016, seawater samples have been most often
directly measured using a wire mesh (liner) to limit the spreading of sea salt in the vaporizer
(https://www.picarro.com/sites/default/files/Salt%20Liner%20App%20Note_180323_final.pd
f).
We most commonly used a Picarro L2130-i CRDS, but at times, a Picarro L2120-i CRDS was
used, resulting in a larger standard deviation, in particular for $\delta D$. On both CRDS analyzers,
when repeatability of the different injections of the sample was not sufficient or the daily run
presented a too large drift, the samples were analyzed at least a second time. In that case, either
the best value or an average of the different values was taken/retained.

The typical daily run at LOCEAN currently includes one or two reference water samples
followed by three freshwater standards at the beginning to establish a slope calibration, as well
as regularly interspersed reference water samples afterwards (usually, from KonaDeep mineral
water with a value close to 0.8 / 2.0 ‰ in $\delta^{18}O$ / $\delta D$). In addition to these freshwater in-house
reference materials, a series can contain up to 12 isotopically-uncharacterized water samples,
using a little over 1 ml of the sample placed in a cap-closed vial. Until 2015, when samples
were distilled, series typically included 12 water samples. Since 2015, when salt water was
directly placed in the vials, we have mostly run not more than 9 samples in a run, because the
deposit of salt in the liner induces water retention or release, and thus noise in the measurements
after roughly 60 injections of salty samples, as well as drifts in the reference water (Fig. 1) and
possibly slope calibration. Another source of drift is the appearance of condensation on the top
cap of the vials after a few hours, which will result in enriching the residual vial water, although
it is by no means the largest source of drift.





Each sea water sample is injected 6 times, and usually 9 to 12 times for the internal standards
at the beginning and end of the run. Whenever possible, samples expected to be in the same
range of values are placed together in the run to minimize the memory effect on the CRDS
which is largest for δD. We reject the first injection, as well as later injections if they are not
stable, retaining between two and eight injections that we average. Two methods were tested,
an empirical one, when we look for successive injections of the sample with close values
(typically 0.02‰ in $\delta^{18}$O), and the systematic selection of the values within 1 sigma starting
with the last three injections. The retained injection values are then averaged. Differences in
the estimates produced by the two methods is usually within 0.02 ‰ in $\delta^{18}$O (0.10 ‰ in δD for
the L2103-i). In the current database, the data retained are the ones obtained with the empirical
approach.
If a significant drift in the reference water values is noticed through the run, it is corrected,
usually by adjusting it linearly between the successive values of the reference water (Fig. 1).
We thus assume that the estimated drift is independent of the $\delta^{18}$O, δD values. In addition, in
2017-2019, the response slope of the Picarro CRDS was adjusted by interpolating between the
three-point slope estimate (based on 3 internal standards) at the beginning and at the end of the
runs, when that was deemed possible. However, this adjustment was discontinued in 2020
because the last internal standard samples were often not as reliably measured, with values more
sensitive to the number of injections, probably as a result of salt deposits in the liner. Since
2020, we only check the instrument's response at the end of the run with one of the freshwater
internal standards.
Accuracy is best when samples are distilled, and for δD it is better on the Picarro CRDS L2130-
i than on the Picarro CRDS L2120-i. Usually, the reproducibility of the $\delta^{18}$O measurements
between the different selected injections is within ± 0.05 ‰ and of the δD measurements within
± 0.15 ‰, which should be considered an upper estimate of the random error on a measurement
with the Picarro L2130-i CRDS. Samples with a SD larger than 0.06 ‰ in $\delta^{18}$O were considered
too uncertain and were rerun, as well as often (after 2015) the first and last samples of each run.
In addition to the instrumental error of each sample $\delta^{18}$O and δD described above, other
uncertainties arise from the data processing and conversion of measured $\delta^{18}$O and δD into the
VSMOW scale. These additional sources of uncertainties are detailed in the next sections.
2.2.2 Data processing
The second source of uncertainty (for Picarro CRDS) is due to the way we process the data of
a daily run with salty water samples. As commented above, we first adjust the values to
compensate for the drift in reference water. Usually, this drift is relatively small, not exceeding
0.1/0.6 ‰ in $\delta^{18}$O / δD during the run, but in about 10% of the runs, it exceeded 0.2 ‰ in $\delta^{18}$O
over the whole run, or 0.10 ‰ in $\delta^{18}$O over successive reference water samples (23 out of 214
daily runs over which statistics were established from 06/2020 to 04/2021). When these large
changes are encountered, the run is estimated noisy and is usually rerun. However, even for the
other runs, a drift is usually observed with salty samples, and it often is a positive drift, in
particular between the reference water samples before and after the three initial internal
standards (Fig. 1). The average (SD) drift in reference water during a run was 0.081 (0.106)
‰ in $\delta^{18}$O, and 0.62 (0.53) ‰ in δD in the 191 (out of 214) daily runs retained. The drift is
also found in the internal standard water analysed at the end of the run compared with the one
analysed just after the initial reference waters with an average (SD) drift of 0.069 (0.073) ‰ in



$\delta^{18}O$, and 0.43 (0.34) ‰ in $\delta D$ for the same 191 daily runs subset. These values slightly differ
from the drifts for the reference water, not significantly at 99% for $\delta^{18}O$, but significantly
at 99% for $\delta D$. This may be indicative of errors resulting from linearly adjusting the drift, in
particular for the initial standard water samples. This suspicion of a slight non-linearity in the
initial drift is reinforced by 7 runs in 2020-2021 when the three standards were also measured
at the end of the run. However, as this is too uncertain, a correction has not been attempted for
that, but in addition to being a source of random error (at least 0.02/0.1 ‰ in $\delta^{18}O$ / $\delta D$) for
individual runs, this might also contribute to absolute errors (i.e. in the VSMOW scale) in the
range of 0.01/0.05 in $\delta^{18}O$ / $\delta D$.
Occasionally, after the correction of the drift, the value of the last internal standard (last sample
port of the run) is shifted for no obvious reason, sometimes by more than 0.10‰ in $\delta^{18}O$ from
what is expected. This might result from a temporary pollution that influences the
measurements (organic matter or particles, either left in the cavity of the vaporizer, on the filter
or on the salt liner), which can also happen for other sample ports. Often, when this happens,
there is also a larger scatter between the different injections, either for this sample or the initial
in-house standards. Running the set of samples again or a selection of them, sometimes
evidences isotopic shifts that can exceed 0.05/0.2 ‰ in $\delta^{18}O$ / $\delta D$. Repeating the analysis helps
mitigate this source of uncertainty. But, this has not always been done, except for data sets on
which there was a specific emphasis.
2.3 Internal standard waters
The last large source of uncertainty is the value (in the VSMOW scale) attributed to the internal
standards used. On the Isoprime IRMS, most internal standards were extracted from different
batches of 'Eau de Paris' (EDP) stored in a tank covered with paraffin, whereas since 2012,
three internal standards are regularly extracted from metal tanks where they are kept for up to
5-6 years with a slight overpressure of dry air (following Gröning, 2018, TEL Technical Note
No. 03). The internal standards have been calibrated using VSMOW and GISP (or GRESP),
usually more than once, and some were also sent to other laboratories at different times to
independently evaluate their characteristics. Comparisons were done in 2013-2014 for three
internal LOCEAN standards with 6 laboratories for $\delta^{18}O$ and 4 laboratories for $\delta D$, which, taken
together, did not reveal an average bias larger than 0.01/0.10 ‰. However, there seems to be
differences for the individual standards (Table 1), with the one at -3.26/-21.32 ‰ presenting an
average positive difference of 0.029/0.19 ‰, whereas the two other ones present a negative
difference (i.e. LOCEAN standards seemed too low) smaller or equal to -0.01/-0.19 ‰.
After further limited comparisons in 2017-2018, that were not conclusive and mostly internal,
the next round of comparisons of the LOCEAN internal standards took place in 2019-2021,
with 5 other European laboratories and for two of them, two different setups for $\delta^{18}O$ (most of
those with IRMS, except for one with a PICARRO L2130 CRDS). Thus, this includes 7
comparisons for $\delta^{18}O$ and 5 for $\delta D$. This set of comparisons (Table 1) was done for the three
internal standards used in 2019-2021, and presents a large scatter between the different
laboratories, on the order of 0.055/0.7 ‰ in $\delta^{18}O$ / $\delta D$. As the differences between laboratories
are similar for the three internal standards, this suggests some systematic differences between
laboratories. However, the large scatter implies that the average differences found are very
uncertain. The differences found for the three internal standards used in 2019-2021 range in
$\delta^{18}O$ / $\delta D$ between 0.029/0.21 ‰ for the most negative standard to -0.010/0.02 ‰ for the most
positive one (Table 1). This might indicate that we have a positive bias for two of our recent
internal standards. This could also produce a small difference in the response slopes of the



Picarro CRDS adopted since 2020. A set of four calibration runs done in November 2021 at
LOCEAN with new VSMOW, GRESP as well as three USGS standards with intermediate
values confirmed a positive bias on the most negative internal standard (MIX2). This run
however did not confirm the average biases on the other internal standards at LOCEAN
suggested by Table 1, nor any major slope error. Therefore, the correction of a systematic bias
has only been applied on the MIX2 value for analyses since August 2020. For some internal
standards, we witnessed larger differences for measurements done in June 2020 after the
L2130-i just returned from a cruise and long shipping and storage for more than 9 months. We
assume that this anomaly is instrumental, and did not last for a long time, as the anomaly was
not reproduced during later tests in August 2020, nor in November 2021.
The two storage methods used successively for internal standard waters were designed to
minimize water vapor exchange. It is however possible that small isotopic drifts of the internal
standards have taken place with time, due to evaporation or possible oxidation of the tanks (rust
was found in one nearly empty tank). As mentioned, based on different comparisons over time,
sometimes over remnants of the tank waters, we could verify that these drifts have remained
smaller than 0.02/0.1 ‰ in $\delta^{18}O$ / $\delta D$. Finally, standards for the daily runs are temporarily
stored, for up to a month, in glass bottles stored at 4°C, which are briefly opened every day to
extract water. Through its storage life, this water will slightly breath, by exchange with the
outside air that penetrates when the bottle is briefly opened. Back of the envelope estimates
suggest that the effect should be less than 0.01/0.05 ‰ in $\delta^{18}O$ / $\delta D$, even after a month.
2.4 Conversion to the concentration scale
Both oxygen and hydrogen isotope compositions are reported in parts per thousand (‰) on the
VSMOW scale. One issue is that we analyse saline samples on the activity scalewhile the
internal standards are fresh water standards, and the method of analysis has changed over time.
We have adjusted LOCEAN data converting them from the activity to the concentration scale
based on the study of Benetti et al. (2017a) as well as on complementary tests with the different
wire meshes used more recently and between duplicated IRMS/CRDS samples. The values we
report are thus internally consistent, but could present differences with datasets processed in
other institutions without this adjustment of up to 0.10/0.20 ‰ in $\delta^{18}O$ / $\delta D$, as indicated in
Benetti et al. (2017a). For example, Walker et al. (2016) find very close $\delta^{18}O$ values in
unadjusted measurements of the same saline samples done on different IRMS and CRDS
instruments. We thus expect that adjusted LOCEAN CRDS $\delta^{18}O$ data would be higher (more
enriched in heavy isotopes) than these other CRDS and more common IRMS data.
2.5 Correction and flagging of samples having probably breathed
In regions where there is enough information in the LOCEAN dataset to establish an average
relationship between d-excess and salinity (Benetti et al., 2017), a large evaporation of a sample
during storage can be detected using its d-excess value, which is then too low compared to the
expected relationship. This was recently checked on a set of 10 water samples originating from
salinity bottles collected in the surface North Atlantic in 2021 on MV Tukuma Arctica that did
not have the usual plastic insert, and thus had breathed as witnessed by the comparison of their
salinity with thermosalinograph records. These samples indeed present, higher practical salinity
(S), d-excess lower than expected and $\delta^{18}O$ and $\delta D$ higher than the expected values, estimated
by average linear fits of d-excess versus salinity and $\delta^{18}O$ versus S for this region. The average
values of the deviations are $\Delta S$=0.29, $\Delta\delta^{18}O$=0.15‰; $\Delta\delta D$=0.33 ‰, $\Delta$d-excess=-0.82 ‰. The
deviations from these expected values present a loose relationship of the deviation in $\delta^{18}O$
($\Delta\delta^{18}O$) on the order of -20% of the deviation of d-excess ($\Delta$d-excess) (Fig. 2). This relationship
is close to the one used by Benetti et al. (2017b) based on other data in the Labrador Sea, where



$\Delta\delta^{18}O$=-1/7 $\Delta$d-excess, $\Delta\delta D$=2 $\Delta\delta^{18}O$ and $\Delta$d-excess = 0.34 $\Delta S$. On the other hand, the
correlation between $\Delta$d-excess and $\Delta S$ is not significantly different from 0, which might be
caused by uncertainties on sampling time causing errors in estimating salinity deviation.
In cases when breathing was not too large (resulting in an increase of less than 0.11‰ in $\delta^{18}O$),
we used the deviation from the expected d-excess relationship to S to estimate an adjusted $\delta^{18}O$
and $\delta D$ (Benetti et al., 2017b). When this method is used, $\delta^{18}O$ and $\delta$ D data are flagged to
'probably good' and d-excess to probably bad, as these data are certainly not as accurate as the
data with no 'correction', with the adjustment adding an uncertainty on the order of 0.05/0.10
‰ in $\delta^{18}O$ / $\delta D$. For larger suspected evaporation, $\delta^{18}O$ and $\delta D$ data are flagged as 'probably
bad'. Altogether, we have flagged 12.3% of the CRDS-measured samples, most of which
(11.3%) correspond to unadjusted data with anomalously low d-excess and thus suspected
evaporation. There is of course also the possibility that for some samples, too low or too high
(for 1% of the cases) d-excess might just result from an occasional large uncertainty in the
analysis.
We recently tested the effectiveness of applying this adjustment for 32 samples from cruise
OVIDE2018 (North Atlantic Ocean in 2018; Lherminer, 2018) which were from pairs of
samples analyzed both by CRDS at LOCEAN and by IRMS at Geozentrum Erlangen, and out
of which 11 LOCEAN-analyzed samples had been slightly adjusted based on their low d-
excess. An average difference is estimated between the 21 non-adjusted samples at LOCEAN
and the IRMS data, which we apply to all the IRMS data before comparison. The comparison
suggests that the adjustment we applied to some of the LOCEAN data, based on their d-excess,
results in diminishing from 0.060 to 0.041 ‰ the standard deviation of the $\delta^{18}O$ differences
between the 32 LOCEAN and Geozentrum Erlangen isotopic values. The adjustment of the 11
samples also diminished the standard deviation of d-excess from the d-excess versus S
relationship derived for the entire LOCEAN dataset from 0.25 to 0.15 ‰. As a comparison,
when the set is restricted to the 21 non-adjusted LOCEAN samples, the corresponding standard
deviations for the $\delta^{18}O$ differences between LOCEAN and Geozentrum Erlangen values, and
d-excess differences to the expected d-excess versus S relationship were 0.043 and 0.14 ‰,
respectively. These values are very close to what is found for the set of 32 samples including
the 11 adjusted samples, suggesting that we have not over-adjusted the LOCEAN samples.
For earlier IRMS analyses at LOCEAN, we base the identification of possible evaporated data
on excessive scatter in the $\delta^{18}O$ versus S scatter plots or between successive data compared to
what we have previously measured in regions with repeated cruises, and outliers (6%) are
flagged as probably bad. The smaller (by half) proportion of flagged IRMS analyses than for
the CRDS analyses suggests either that this validation missed some evaporated IRMS samples,
or that these earlier data had evaporated less than the more recent ones (some were analyzed
sooner after collection), or that the IRMS runs had smaller uncertainties than the latter CRDS
runs.
3. Validation
As discussed in section 2, in addition to random errors or to issues related with evaporation of
samples, there is the possibility of shifts between subsets of the data, due to the different internal
standard waters, methods of processing or conversion from the activity to the concentration
scale. We thus need to compare this database with data analyzed in other laboratories, and
evaluate time series when the data have been repeated in time at the same location. In particular,
the LOCEAN dataset contains a limited number of samples for different cruises in deep-water
masses that are unlikely to have experienced much change in their isotopic composition over





the last 50 years, due to their weak ventilation and small salinity variability. Examining data in
such deep-waters can thus provide a test of consistency between subsets of the LOCEAN data,
or relative to other datasets.

Within the LOCEAN dataset, relevant deep waters have been sampled in different years (in the
Southern Indian Ocean (OISO cruises), in the equatorial Atlantic (PIRATA cruises) and in the
North Atlantic subpolar gyre (mostly OVIDE cruises), with statistics presented in Table 2.
These comparisons on a limited set of cruises, but corresponding to analyses done throughout
the 22 last years of the spectrometry platform suggest that internally the $\delta^{18}$O dataset is coherent
in time to within 0.03 ‰ (after an adjustment applied on LOCEAN IRMS data which most of
the time was +0.09 ‰ to adjust to CRDS data). For $\delta$D, the period of comparison is more
limited with data from Picarro CRDS only since 2010, and the standard error of yearly $\delta$D
averages is typically on the order of 0.15 ‰. The comparison also highlights cruises with more
noisy data than others. This is for example the case of the 2002 OISO08 IRMS data (without
the OISO08 data, the mean (standard error) $\delta^{18}$O for subset 1 decreases to 0.078 (0.030) ‰).
There are also some suggestions of systematic differences between cruises (for example, for
subsets 1-2, OISO29 (2019) samples tend to have lower $\delta^{18}$O and $\delta$D values, whereas OISO31
(2021) samples tend to have higher values). However, this is within the uncertainties of the
means and is not fully understood. Thus, no further correction is warranted.

There are $\delta^{18}$O data from a few cruises sampling deep-waters which can be compared with
subsets of the LOCEAN data. These together with duplicates sets of samples between
LOCEAN and other facilities form the basis for estimating consistency relative to the other data
(details in App. A). The different comparisons yielded very varied results. It is often difficult
to understand what is the source of the differences, but one commonly suspects choices of
protocols, characteristics of the instrument used or internal standards (see also Aoki et al, 2017;
Wassenaar et al., 2021). Altogether, although the limited inter-comparisons listed above have
a large scatter (the standard deviation in the set of 17 average differences listed in App. A is
0.055 ‰), there is a tendency for LOCEAN $\delta^{18}$O values reported in the concentration scale to
be higher (relatively enriched in heavy isotopes). The average of these 17 different comparisons
is 0.093 ‰ with a standard error of 0.013 ‰. This average difference happens to be close to
the 0.09 ‰ change to the concentration scale that was applied to recent CRDS salty water
samples analysed since 2015 at LOCEAN, an adjustment that is not done on CRDS or IRMS
datasets produced in other facilities.

In summary, these external comparisons, together with the internal consistency tests on the
LOCEAN database in a few regions, suggest that the LOCEAN $\delta^{18}$O dataset are within +0.035
‰ absolute accuracy, at least when averaged spatially or in time (Table 2). Individual data have
larger uncertainties as discussed before, because of the instrumental and internal standards
uncertainty (resulting in a total uncertainty of usually less than 0.05 ‰ in $\delta^{18}$O) and possible
aging/evaporation during collection and storage. We are not able to provide similar
comparisons for $\delta$D or d-excess, as the database for comparison is much reduced.

4. The data
4.1 Data distribution
Fig. 3 presents the spatial distribution of the LOCEAN-analyzed data close to the surface, with
the largest data collection being in the North Atlantic (in particular, with OVIDE cruises since
2002 and the SURATLANT ship of opportunity dataset since 2011), the tropical Atlantic (in
particular, the EGEE and PIRATA cruises since 2005), and the South Indian Ocean (OISO
cruises since 1998).





Table 3 reports the number of valid data points by depth range, which indicates that the
emphasis in this set has been on near surface data (58% of the $\delta^{18}$O data above 40m depth, with
13% between 40 and 200m depth, and only 12% at 1000m or deeper). There is less valid $\delta$D
than $\delta^{18}$O data, the difference corresponding to IRMS-measured data, which correspond to 25%
of the total number of water samples in the database. There is even less valid d-excess than $\delta$D
(by 10%), the difference corresponding to samples for which an adjustment for slight
evaporation was done on $\delta^{18}$O and $\delta$D data. The database contains fewer deep samples since
the transition to CRDS than before, because of a recent emphasis of sampling the upper ocean.
4.2 Time series
We illustrate the dataset with time series of June (or July) data between 50° and 55°N in the
eastern North Atlantic subpolar gyre (NASPG) collected mostly during the OVIDE cruises
(Fig. 4). This scatter plot of cruise-averaged S and $\delta^{18}$O indicates a near alignment of the values.
It is striking that the strongest negative (fresher/lighter) anomalies in 2016 fit rather well on the
regression line (in red) for water samples in the southwestern NASPG. This regression line is
derived from data from the 47–55°N, 30-49°W region, excluding very low salinity data from
seasonal sea ice melt or from shelf waters, and is very similar to the distribution in Frew et al.
(2000). Thus, this reinforces the hypothesis of Holliday et al. (2019) that the strong freshening
present in the eastern subpolar gyre in 2016 originated from the transport of Arctic freshwater
from the western boundary current into the eastern basins, and not from local rainfall, which
would have likely resulted in higher $\delta^{18}$O at the same 'low' salinity such as depicted by the
black line (Frew et al., 2000; C. Risi, pers. comm., 2021).
The SURATLANT surveys provided a seasonal sampling of water isotopes between late 2011
and 2019 along the western flank of the Reykjanes Ridge in the central part of the gyre (53-
56°N/38°-44°W). Annual summaries of these data are provided on Fig. 5a. There is less
alignment of the interannual values on the average southwestern NASPG linear regression line
than for the OVIDE surveys (Fig. 4). However, there is some aliasing of the seasonal cycle in
the annual averages (see Reverdin et al., 2018b), which contributes to the scatter, as well as
noise on the data, and natural variability. On this plot the freshest year appears to be 2017, in
agreement with an analysis using a much more complete salinity dataset (Reverdin et al.,
2018a). 2017 is also one of the lighter $\delta^{18}$O years. The corresponding d-excess versus S diagram
(Fig. 5b) presents yearly anomalies that are fairly aligned with the average regression between
southwestern NASPG d-excess and salinity data. Error bars are large, but nevertheless, low
salinity waters exhibit high d-excess, as described in Benetti et al. (2016).
5. Data availability:
The dataset described is version V2 at https://doi.org/10.17882/71186 (Waterisotopes-
CISE-LOCEAN, 2021).
6. Conclusions
Instrumental uncertainty on individual data in this dataset is as low as 0.03/0.15‰ for most
runs, with occasional much larger uncertainties. One needs to add to that the uncertainties on
the internal standards that are used to convert measured values into the VSMOW scale.
Different comparisons suggest that the internal standard values have almost always remained
defined within at most 0.03/0.2‰ for $\delta^{18}$O / $\delta$D since 2012. There was however a short-term
larger difference found for the most negative standard (equal to 0.1‰ for $\delta^{18}$O ), most likely
related to the readjustment of the instrument to laboratory conditions in May 2021. When using
the CRDS Picarro L2130-i, we also found periods with quite uncertain analyses, in particular



due to salt or particle deposit in the vaporizer or filters. These samples could often be run again
afterwards to reach lower resulting uncertainty.

Finally, there is the issue of possible evaporation during collection and storage. When the
analysis is done on a CRDS, we are usually able to detect possible biases larger than 0.05‰ in
$\delta^{18}O$, by comparing d-excess with the expected d-excess derived from regional d-excess-S
linear relationships. Attempts were made here to correct $\delta^{18}O$ and $\delta D$ when the resulting
uncertainty does not exceed 0.05/0.1‰. In particular this was done for some OISO cruise
samples which were analysed many years after collection, or in the case of faulty caps being
used, or caps that were not properly closed and with no parafilm. This is certainly a strong
source of quality loss for part of the database, and 'small' effects may have remained
undetected.

Possible long-term drifts due to changes in internal standards, storage, instrumentation and
protocols are difficult to estimate. This is done here by checking the consistency of different
subsets of the database, for instance when time series can be obtained (such as in the southern
Indian Ocean or North Atlantic subpolar gyre), or by comparison with duplicate data analysed
in other laboratories, or with other datasets in deep regions commonly sampled. These
comparisons are encouraging. On one hand, they suggest that the internal consistency in the
database is usually within a 0.03/0.15‰ uncertainty for $\delta^{18}O/\delta D$. On the other hand, although
other datasets sometimes differ by much more with a large scatter between the 17 comparisons
(with a standard deviation of 0.055‰ for $\delta^{18}O$), the average difference (0.093‰) found with
them is close to the change that is applied to the LOCEAN data to report them on the
concentration scale (0.09‰ for $\delta^{18}O$ analyzed with a salt liner since 2015). Of course, there is
still the possibility of errors and biases in subsets that could not be compared in a similar way,
such as surface samples collected from ships of opportunity or sailing vessels in the tropics,
that could result from different handling of the samples during collection and more uncertain
storage conditions. There are also small errors originating from memory effects in the Picarro
CRDS runs that could be better corrected and taken into account (Vallet-Coulomb et al., 2021).

We also illustrated the possibility of using this dataset to investigate ocean variability. Of
course, the interest of a data archive is to merge different institutes datasets such as this one,
while retaining a similar accuracy. This was attempted in the Global Seawater Oxygen-18
Database at GISS (Schmidt et al., 1999), although biases between subsets of this mostly $\delta^{18}O$
dataset remain at a level that makes the overall analysis of variability difficult to carry. The few
comparisons we could do suggest that differences with other datasets are at times large. The
effort to correctly adjust for these differences and produce a larger coherent archive is required
to get full use of the data collected. There is still a need of more and better calibrated sea water
isotope data to reconstruct tropical hydroclimate variability, such as formulated for the tropical
coral archives by PAGES CoralHydro2k Project, or for high latitude studies of the various
sources of freshwater in the ocean, including continental runoff, sea ice, iceberg melt and air-
sea exchanges.

Appendix A: Comparisons of LOCEAN data with other isotopic data
This includes on one hand comparisons with data of other cruises, in areas where we expect
variability to have been weak, such as in the deep ocean, and on the other hand, considering
duplicate sets of samples analysed in different institution.

Akhoudas et al. (2021) used the first approach in the deep Weddell Sea, comparing the
LOCEAN 2017 Wapiti cruise data with data from other cruises over a fairly large range of



neutral density surfaces. They identified a cruise whose $\delta^{18}O$ values were lower by 0.13‰ than
at LOCEAN, as well as datasets that fit the Wapiti cruise values to within the data uncertainties
(for example, from ANT-X12 cruise on RV Polarstern in 1995). Another water mass which can
be used for comparison is the near - bottom waters in Fram Strait (below 2000m), which are
either originating from the Arctic Ocean, or recirculating from the Greenland Sea. This water
mass is regularly sampled, and has not been strongly ventilated recently. In 1998-2015 during
German-led cruises, these waters presented an average $\delta^{18}O$ value close to 0.28‰ (after
removing suspiciously high data of a cruise in 2011 and large positive outliers in 2012; Paul
Dodd, personal communication). The LOCEAN database contains seven $\delta^{18}O$ samples close to
the bottom across Fram Strait from MSM76 cruise on RV Maria S Merian in 2018, with average
(SD) value close to 0.395 (0.035) ‰, thus averaging higher by 0.115‰.
We extracted individual profiles from the GISS Global Seawater Oxygen-18 Database
(Schmidt et al., 1999) that can be compared with the LOCEAN station data, in deep and old
water masses. In the southern Indian Ocean, for example numerous profiles collected during
1993-1994 cruises (CIVA1 (Archambeau et al., 1998), ADOX1, SWINDEX, ADOX2)
suggest that LOCEAN $\delta^{18}O$ in the deep layers are higher by 0.10 to 0.17 ‰ depending on the
cruise. There is also one GEOSECS 1978 station with a single deep value within 0.01 ‰ of
close-by OISO stations, as well as some 1984 (INDIVAT1) and 1996 (CIVA2) station data
with larger uncertainties that indicate higher LOCEAN $\delta^{18}O$ values by 0.15 to 0.22 ‰,
depending on how outliers are identified and removed.
In the North Atlantic, there are data from three cruises that can be directly compared with
LOCEAN data, focusing on deep waters with T-S properties close to the ones of the
LOCEAN dataset. Comparison with one GEOSECS 1972 station south of Greenland suggests
higher $\delta^{18}O$ LOCEAN values by ~ 0.060 ‰ (there is a small salinity shift between the two
profiles which required to adjust the LOCEAN $\delta^{18}O$ value to the same salinity based on the
average $\delta^{18}O$-S relationship). Data of 4 stations of the CONVEX1991 cruise (Frew et al.,
2000) indicate higher $\delta^{18}O$ in LOCEAN dataset by ~ 0.090 ‰ (after adjustment done to
consider small salinity differences). On the other hand, data close to the North East Atlantic
deep-water layer from stations collected in 6/1995 in the southern Labrador Sea (Khatiwala et
al., 1999) do not show a significant difference with LOCEAN stations closer to south
Greenland (southern Irminger Sea) at a similar salinity. In the equatorial Atlantic there are
deep data of two GEOSECS stations collected in 10/1972 and 2/1973 that can be compared
with the LOCEAN data (mostly near 1000-2000m depth). These limited comparisons (often
at large distance, but at a similar salinity) suggest that LOCEAN values are larger than the
GEOSECS $\delta^{18}O$ by 0.055 ‰.
Finally, there are a few instances of sea water samples that have been duplicated and shared
with other laboratories. Some of these in 2013-2014 have been used to validate how to convert
IRMS or CRDS measurements into the concentration scale, with or without distillation (Benetti
et al, 2017), that we will not include here, and that suggested a scatter in the comparisons with
different IRMS laboratories for natural or artificial sea water samples often on the order of 0.10
‰. More recently, 18 samples of the WAPITI2017 cruise were duplicated with analyses both
at LOCEAN and at the British Geological Survey stable isotope facility (BGS), which indicated
lower LOCEAN $\delta^{18}O$ averaging -0.09 ‰ (SD = 0.035 ‰) (Akhoudas et al., 2021). In the same
region, a small set of 11 samples was duplicated in 2020 with Hokkaido University, which
suggests that LOCEAN $\delta^{18}O$ values are higher by 0.139 ‰ with a SD of 0.019 ‰ (Shigeru
Aoki, pers. comm., 2021).



There have also been duplicates of LOCEAN samples during OVIDE cruises in 2010, 2016 and
2018 analysed in different facilities (Antje Voelker, pers. comm., 2021), which suggested
different average differences for the different years. In particular for 2016 samples close to
2500m, LOCEAN values average higher by 0.035 ‰, whereas in 2018, the average difference
is closer to 0.07‰, but with a few stations at the north-western end of the section in Irminger
sea with differences on the order of 0.02 ‰.
Author contribution:
Gilles Reverdin and Claire Waelbroeck have measured parts of the isotopic data,
contributed to their validation and written the paper.  Catherine Pierre, Camille
Akhoudas, Giovanni Aloisi, Marion Benetti, have measured parts of the isotopic data and
contributed to their validation. Jérôme Demange has maintained the CISE-LOCEAN IRMS
and CRDS and Aïcha Naamar has measured parts of the isotopic data. Denis Diverrès,
Magnus Danielsen and Thierry Reynaud have contributed water samples from ships of
opportunity with associated salinity measurements. Bernard Bourlès, Jean-Claude
Gascard, Hervé Le Goff, Marie-Noëlle Houssais, Pascale Lherminier, Claire Lo Monaco,
Herlé Mercier, Nicolas Metzl, Simon Morisset, Jean-Baptiste Sallée, Virginie Thierry, Susan
Hartman, Ed Mawji, Solveig Olafsdottir, Torsten Kanzow, Antje Voelker, and Igor
Yashayaev have contributed to the sample collection, and in some cases provided
duplicate samples from other institutions.
Competing interests:
The authors declare that they have no conflict of interest.

Acknowledgments: Data were measured at the CISE-LOCEAN facility housed by the
LOCEAN laboratory and part of the OSU ECCE Terra analytical services. Support by OSU
ECCE Terra, by LOCEAN, and by various French national institutes and programs is gratefully
acknowledged (including INSU/CNRS, IFREMER, IRD, IPEV, LEFE program, ANR
GEOVIDE), as well as support by different French 'Services nationaux d'Observation', such
as PIRATA, SSS and OISO/CARAUS. Many of the data originate from research cruises on
French Research vessels: R.V. Suroit, Thalassa, Atalante, Marion Dufresne 2, Tara. Some data
were collected during research cruises on non-French vessels, such as MIDAS in 2013 and
BOCATS in 2016 on the Spanish R.V. Sarmiento de Gamboa, HUD2014007 on the Canadian
R.V. Hudson, 2014 JR302 in 2014 and 2017 JR16004 cruises on the U.K. HMS James Clarke
Ross, the Arctic cruises in 2006-2008, 2013, and the 2020-2021 Microbiome cruise on French
S.V. Tara, the Nordic seas MIZEX cruises in 2002-2004 on Swedish R.V. Oden, the 2017
SPURS2 cruise on R.V. Revelle, and the 2018 east Greenland cruise MSM76 on German R.V.
Maria S. Merian. The SURATLANT data were collected on merchant vessels with support
from Iceland-based EIMSKIP and Marine and Fisheries Research Institute in Iceland and the
Nuka Arctica/Tukuma Arctica dataset was collected on the merchant vessels Nuka Arctica and
Tukuma Arctica from Greenland-based company RAL. Data were also collected from different
Merchant Vessels recruited by SNO SSS for the Atlantic Ocean monitoring. Finally, data were
collected from sailing vessels, including the Rara Avis (AJD), the Boogaloo and Ragnar
(OceanoScientific), the Northabout (UnoMundo) and the UltimIII (SODEBO). The WAPITI
project received funding from the European Research Council (ERC) under the European
Union's Horizon 2020 research and innovation program (grant agreement 637770).
Intercomparisons of samples were done with various other institutions to which we are very
grateful. In particular, we acknowledge the contributions by Robert van Geldern at Geozentrum
Nordbayern, Mélanie Leng at the British Geological Survey, Arne E Sveinbjörnsdóttir and Rosa
Ólafsdóttir at the University of Reykjavik, Pal Morkved at the University of Bergen, Bénédicte





Minster at LSCE, Penny Holliday at NOC in Southampton, Paul Dodd at the Norwegian Polar
Institute in Tromsoe, and Shigeru Aoki at Hokkaido University.





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




Table 1
Comparison of standards measured at LOCEAN and in other laboratories (in ‰).

| Date | Internal Standard | LOCEAN $\delta^{18}O$ ‰ | LOCEAN $\delta D$ ‰ | $\delta^{18}O$ deviation ‰ | Nber of $\delta^{18}O$ lab settings | $\delta D$ deviation ‰ | Nber of $\delta D$ lab settings |
|---|---|---|---|---|---|---|---|
| 2013-2014 | EDP | -6.610 | -44.30 | -0.010 | 6 | -0.19 | 4 |
| 2013-2014 | MIX | -3.260 | -21.32 | 0.029 | 6 | 0.19 | 4 |
| 2013-2014 | KONA | -0.050 | 0.46 | -0.007 | 6 | -0.18 | 4 |
| 2019-2021 | MIX2 | -2.610 | -17.93 | 0.029 | 7 | 0.21 | 5 |
| 2019-2021 | BERING | -0.805 | -4.56 | 0.028 | 7 | 0.19 | 5 |
| 2019-2021 | KONA3 | 1.220 | 3.40 | -0.010 | 7 | 0.02 | 5 |


Table 2:
Comparison of LOCEAN annually-averaged data in a few selected deep-water masses which
exhibit little variability in their salinity, and have likely not been recently ventilated:
1: OISO cruises (1998 to 2021) near 1000-1500m in South Indian Ocean Antarctic sector of
the Southern Ocean (50°S-58°S) (1998*, 2002*, and most years since 2010)
2: OISO cruises (1998 to 2021) near 2000m in the western South Indian Ocean subtropical gyre
(1998*, 2002*, and most years since 2010)
3: PIRATA and EGEE cruises (2005-2021) near 1000m in eastern equatorial Atlantic (2005*,
2006*, 2007*, 2015, 2020, 2021)
4: OVIDE and RREX2017 data between 2000m and 3500m in eastern North Atlantic subpolar
gyre (data in 2002*, 2016, 2017, 2018, 2021)
Reported S, $\delta^{18}O$, $\delta D$ and d-excess values are average values for all samples and all years
(standard error). The number of years (N years) refers to the $\delta^{18}O$ data. The standard error is
computed as the standard error of the different annual averages, i.e. the standard deviation of
the different annual averages divided by $\sqrt{N}$.

| Cruise set | 1 | 2 | 3 | 4 |
|---|---|---|---|---|
| N years | 13 | 9 | 6 | 5 |
| S | 34.710 (0.005) | 34.695 (0.002) | 34.615 (0.005) | 34.936 (0.005) |
| $\delta^{18}O$ (‰) | 0.095 (0.035) | 0.085 (0.025) | 0.150 (0.009) | 0.287 (0.025) |
| $\delta D$ (‰) | -0.25 (0.13) | -0.29 (0.09) | 0.24 (0.14)** | 1.18 (0.18) |
| d-excess (‰) | -0.80 (0.15) | -1.03 (0.18) | -0.81 (0.0)** | -1.05 (0.09) |

* IRMS estimates for $\delta^{18}O$ only.
** only two years

Table 3: number of valid sea water isotopic data by depth range in Waterisotopes-CISE-
LOCEAN (2021, version V2) (a total of 7595 valid data for $\delta^{18}O$ out of 7703 data entries)

| Depth range (m) | $\delta^{18}O$ (‰) | $\delta D$ (‰) | d-excess (‰) |
|---|---|---|---|
| 0-40 | 4517 | 3416 | 3180 |
| 40-199 | 1029 | 716 | 625 |
| 200-999 | 1245 | 1029 | 919 |
| > 999 | 804 | 539 | 465 |
| total | 7595 | 5700 | 5189 |




Figure captions
Figure 1: A typical run (on 2/08 2021) of 19 samples using three internal standards and
KonaDeep-water samples (left for $\delta^{18}$O and right for $\delta$D). Top panels: the deviations of isotopic
values (‰) of internal standards (in blue) and of the KonaDeep-water samples (in red) relative
to their expected values. Error bars are the standard deviation of the different injections, and
the vertical scale is arbitrary set so that 0 corresponds to KonaDeep sample 6 (after the three
internal standards). The lower panels present the values obtained after adjusting for the drifts
identified with the KonaDeep-water samples through the run.
Figure 2: Scatter diagram of the deviation of $\delta^{18}$O (‰) versus the deviation of d-excess (‰) for
a set of samples extracted from salinity bottles with no plastic inserts that had evaporated (2021,
mostly from MV Tukuma in the North Atlantic). The deviations are estimated by subtracting
from the isotopic data the isotopic value estimated as a function of practical salinity, based on
the other regional data. The error bars on each sample are the standard deviation between the
different injections and assuming that the standard deviation of $\delta^{18}$O and $\delta$D are independent
when estimating d-excess. The red line is the regression used in Benetti et al. (2016).
Figure 3: Six maps which include most of the near-surface $\delta^{18}$O data in the LOCEAN archive
(color scale in ‰).
Figure 4: Scatter plot of cruise averages of near surface (upper 100-m) $\delta^{18}$O (‰) versus
practical salinity in the Iceland Basin, close to the NAC fronts. The bars indicate the standard
deviation between the individual data that are averaged. Notice the fresher and isotopically
lighter data from the BOCATS (OVIDE transect) cruise in 2016. The red line corresponds to
the average linear relationship in the south-western NA SPG (SURATLANT dataset within 47–
55°N and 30-49°W, with practical salinity between 33.1 and 35.5), whereas the black line
reports the slope expected from mixing with local rainfall end-member.
Figure 5: Scatter plots in the southern Irminger Sea/NASPG of annually averaged
SURATLANT surveys data. The left panel presents $\delta^{18}$O (‰) versus practical salinity, whereas
the right panel presents d-excess (‰) versus practical salinity. The bars indicate the standard
deviation between the individual data that are averaged. The red lines correspond to the average
linear relationships in the SURATLANT dataset within 47–55°N and 30-49°W, with salinity
between 33.1 and 35.5 (see Reverdin et al., 2018b), the red line on the left panel, being the same
as on Fig. 4.

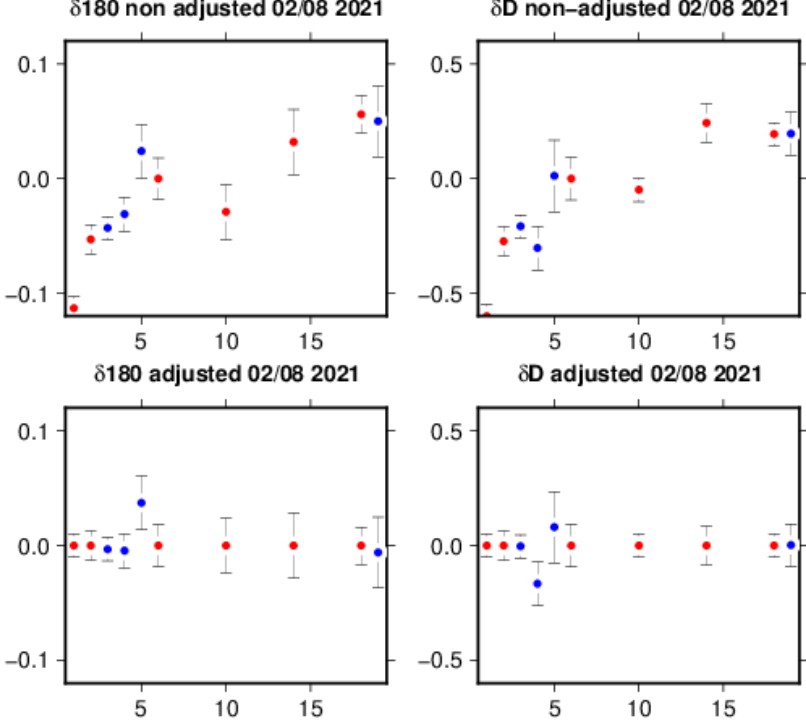

Figure 1: A typical run (on 2/08 2021) of 19 samples using three internal standards and
KonaDeep-water samples (left for $\delta^{18}O$ and right for $\delta D$). Top panels: the deviations of
isotopic values (‰) of internal standards (in blue) and of the KonaDeep-water samples (in
red) relative to their expected values. Error bars are the standard deviation of the different
injections, and the vertical scale is arbitrary set so that 0 corresponds to KonaDeep sample 6
(after the three internal standards). The lower panels present the values obtained after
adjusting for the drifts identified with the KonaDeep-water samples through the run.



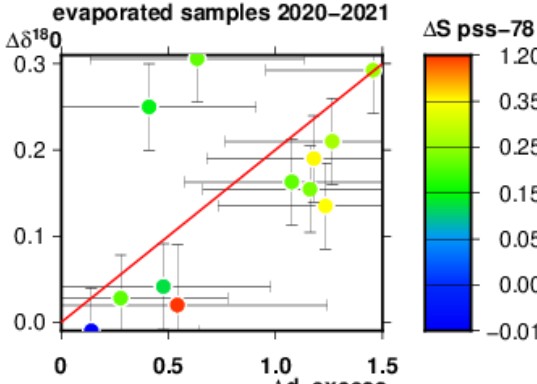

Figure 2: Scatter diagram of the deviation of $\delta^{18}O$ (‰) versus the deviation of d-excess (‰) for
a set of samples extracted from salinity bottles with no plastic inserts that had evaporated (2021,
mostly from MV Tukuma Arctica in the North Atlantic). The deviations are estimated by
subtracting from the isotopic data the isotopic value estimated as a function of practical salinity,
based on the other regional data. The error bars on each sample are the standard deviation
between the different injections and assuming that the standard deviation of $\delta^{18}O$ and $\delta D$ are
independent when estimating d-excess. The red line is the regression used in Benetti et al.
815   (2016).



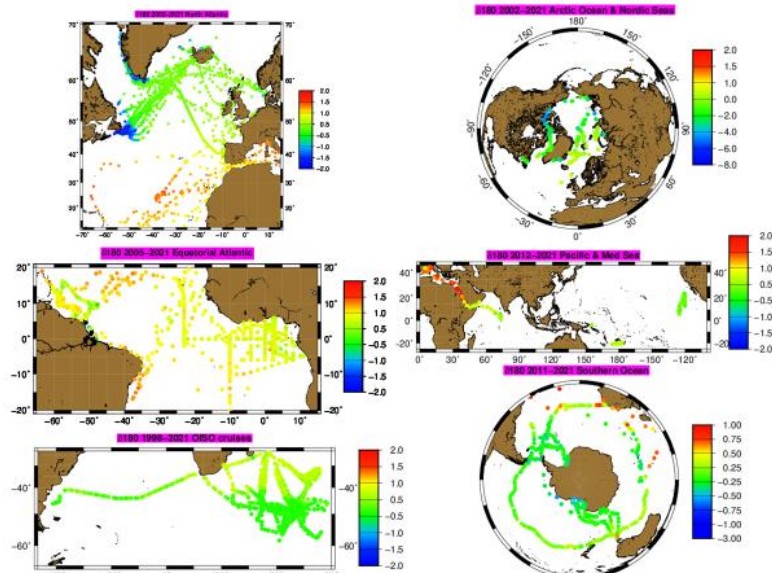

Figure 3: Six maps which include most of the near-surface δ¹⁸O data in the LOCEAN archive
(color scale in ‰).

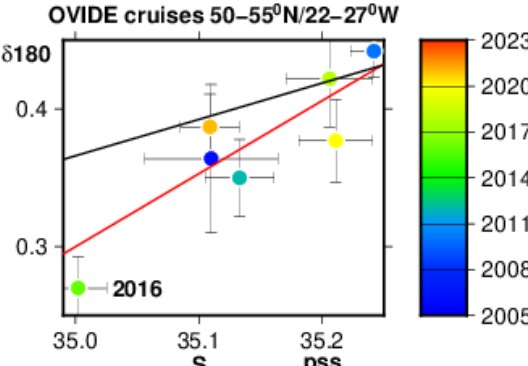

Figure 4: Scatter plot of cruise averages of near surface (upper 100-m) $\delta^{18}O$ (‰) versus
practical salinity in the Iceland Basin, close to the NAC fronts. The bars indicate the standard
deviation between the individual data that are averaged. Notice the fresher and isotopically
lighter data from the BOCATS (OVIDE transect) cruise in 2016. The red line corresponds to
the average linear relationship in the south-western NA SPG (SURATLANT dataset within
47–55°N and 30-49°W, with practical salinity between 33.1 and 35.5), whereas the black line
reports the slope expected from mixing with local rainfall end-member.



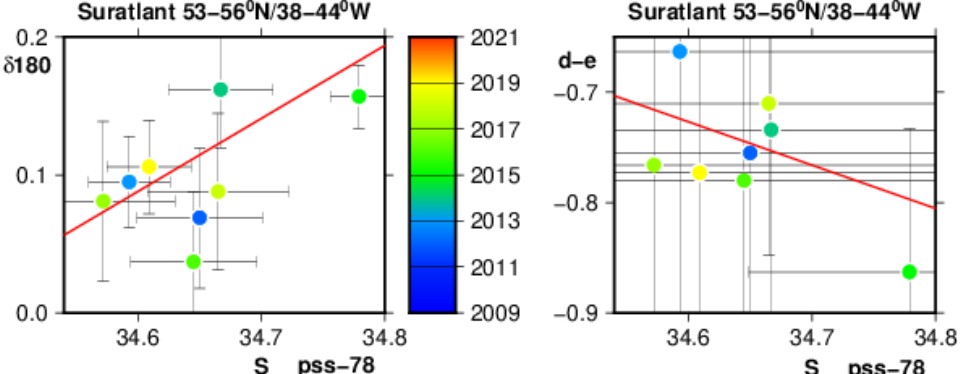

Figure 5: Scatter plots in the southern Irminger Sea/NASPG of annually averaged
SURATLANT surveys data. The left panel presents $\delta^{18}$O (‰) versus practical salinity, whereas
the right panel presents d-excess (‰) versus practical salinity. The bars indicate the standard
deviation between the individual data that are averaged. The red lines correspond to the average
linear relationships in the SURATLANT dataset within 47–55°N and 30-49°W, with salinity
between 33.1 and 35.5 (see Reverdin et al., 2018b), the red line on the left panel, being the same
as on Fig. 4.