# Peer review of "The CISE-LOCEAN seawater isotopic database (1998-2021)"

_Earth System Science Data, 2022_

## Referee Comment (RC1)

**Manuscript ID: ESSD-2022-34**
*„The CISE-LOCEAN sea water isotopic database (1998-2021)"*

**Comments to the authors**

*General remarks*

The manuscript describes the methods and analytical techniques including adjustments and correction applied to seawater stable isotope data ($\delta^{18}O$, $\delta^2H$) in the LOCEAN data base, which hold more than 20 years of measurements from various cruises. The authors discuss the different sources of error and uncertainty. Overall, for $\delta^2H$ the authors are bit too optimistic for the precision of the data. Especially IRMS measurements in the past have not reached the precision of the latest CRDS generation. So, the claimed standard deviation of 0.15 (line 8) is most probably far to optimistic. The data set itself is of high value for various research disciplines and the authors did their best to describe the quality and limitations of the data. I made some more specific comments to some sections below.

Technically, the manuscript is adequality organized and language does not need any revision. With some minor revisions, the manuscript should be publishable in ESSD.

*Specific comments*

L51     Please give the full meaning of the acronym as footnote

Section 2.1     Does the potentially evaporated samples show a deviation similar to an 'evaporation line' in terrestrial surface water, aka the samples plot on a regression with lower slope assuming that the starting value is roughly comparable for a set of samples?

L178-180     From personal experience, I doubt that in-vial evaporation causes drift in the instrument. In such a case, the effect would also be visible in the same manner also by IRMS analysis that uses for example 12mL Exetainers. This is definitely not the case. Thus, possible but very unlikely.

L268-290     According to Table 1 the most negative internal standard has a $\delta^{18}O$ value of -6.61permil, the highest +1.22permil. With respect to the VSMOW-SLAP scale, this is a rather small coverage of the range ($\Delta\delta_{VSMOW-SLAP}$ = 55 permil). Yes, seawater has a rather narrow range of isotope values compared to continental freshwaters or precipitation, however, using reference materials so close to each other will not improve the overall precision of the measurement. For scale normalization you calculate a regression line through 2 or 3 of your standards and if isotope values are rather close any scatter in the 2 to 8 values used for the average calculation will already shift your slope and intercept of the line. This effect should be less problematic for reference materials that show a wider separation in isotope values. Also, the other laboratories might have used a different range of reference materials. Seawater isotope analysis is somewhat different to groundwater or precipitation, which covers a far wider range of isotope values (and the respective standards).

L312     The "salt isotope effect" might need a deeper discussion here. First, this effect only influences IRMS measurements done by equilibration ($CO_2$ and $H_2$) but does not apply to LAS (Picarro, Los Gatos). So, theoretically laser measurements should not be corrected in any manner. This not so clear from the text. You apply a correction approach by Benetti et al (2017) who claim

that also LAS $\delta^{18}O$ data (at least Picarro) should be corrected by 0.09‰ due to incomplete evaporation in the vaporizer.

Second, the effect of seawater salinity to the chemical equilibrium (activity vs concentration scale) has been tested and discussed in quite a few publications. However, studies were never 100% conclusive. This correction is reported to be –0.15‰ for $\delta^{18}O$ (Lécuyer et al, 2009) and -2‰ for $\delta D$ (Martineau et al. 2012) between the fresh water reference materials and the saline samples. However, other studies did not observe such an effect for seawater salt concentrations (Horita et al., 1993a,b; Bourg et al., 2001). Consequently, it is mainly unclear for most datasets if or if not a correction has been applied to the final measured value. Is this stated for the LOCEAN data in the database what correction have been applied to older IRMS data? This is also not really clear from the text.

Bourg, C., Stievenard, M. and Jouzel, J. (2000), Hydrogen and oxygen isotopic composition of aqueous salt solutions by gas–water equilibration method. - *Chemical Geology*, 173, 331-337.

Horita J., Cole D. R., and Wesolowski D. J. (1993a) The activity-composition relationship of oxygen and hydrogen isotopes in aqueous salt solutions: I. I. Vapor-liquid water equilibration of single salt solutions from 50 to 100°. - *Geochimica Cosmochimica Acta,* 57(19), 2797-2817

Horita J., Cole D. R., and Wesolowski D. J. (1993b) The activity-composition relationship of oxygen and hydrogen isotopes in aqueous salt solutions: II. Vapor-liquid water equilibration of mixed salt solutions from 50 to 100°C and geochemical implications. - *Geochimica Cosmochimica Acta* 57(19), 4703-4711

Lécuyer, C., Gardien, V., Rigaudier, T., Fourel, F., Martineau, F. and Cros, A. (2009): Oxygen isotope fractionation and equilibration kinetics between CO2 and H2O as a function of salinity of aqueous solutions. - *Chemical Geology*, 264, 122-126.

Martineau et al. (2012). D/H equilibrium fractionation between $H_2O$ and $H_2$ as a function of the salinity of aqueous solutions. Chemical Geology 291 (2012) 236–240

L340        Has the correction been applied also to the samples with suspected larger evaporation?

L347-362    If the samples were identical, both measurement techniques (LAS and IRMS) should result in the same values as samples has undergone evaporation and were split into aliquots later? I cannot directly see how this then helps to test the evaporation correction approach based on the relation between S, d-excess and δ-values. Or does this paragraph refer to the salt isotope effect above?

*Technical comments*

L312        found

L337        space character between delta and D

---

## Author Comment (AC1)

**Reviewer 1:**
**Comments to the authors**
*General remarks*
*The manuscript describes the methods and analytical techniques including adjustments and correction applied to seawater stable isotope data ($d_{18}O$, $d_2H$) in the LOCEAN data base, which hold more than 20 years of measurements from various cruises. The authors discuss the different sources of error and uncertainty. Overall, for $d_2H$ the authors are bit too optimistic*
*for the precision of the data. Especially IRMS measurements in the past have not reached the precision of the latest CRDS generation. So, the claimed standard deviation of 0.15 (line 8) is most probably far too optimistic. The data set itself is of high value for various research disciplines and the authors did their best to describe the quality and limitations of the data. I made some more specific comments to some sections below. Technically, the manuscript is adequality organized and language does not need any revision.*
*With some minor revisions, the manuscript should be publishable in ESSD.*

**Au**: Thank-you for your comments and appreciation. The 0.15‰ standard deviation for δD refers to the instrumental CRDS uncertainty. We fully agree that it would be much higher for earlier IRMS measurements. The LOCEAN database does not contain any IRMS δD measurements (unfortunately, as the authorization for the installation on the roof was never granted by the university which houses the instrumentation).

*Specific comments*
*L51 Please give the full meaning of the acronym as footnote*
**Au**: CISE is a name initially given to the stable isotopic facility of our university geophysics department ('Centre des isotopes stables environnementaux'). LOCEAN is the name of our laboratory ('Laboratoire d'Océanographie et du Climat'). CISE-LOCEAN refers to the part of the stable isotopic facility housed at LOCEAN. It is a bit too long to be explained. Instead we wrote: 'Since 1998, the isotopic platform facility at LOCEAN (later 'CISE-LOCEAN')…

*Section 2.1 Does the potentially evaporated samples show a deviation similar to an 'evaporation line' in terrestrial surface water, aka the samples plot on a regression with lower slope assuming that the starting value is roughly comparable for a set of samples?*
**Au**: yes, this is a lower slope than the close to 8 classical slope in meteoric water (Rayleigh equilibration) , assuming that the starting value is roughly the same. Typically, the slope seems to be on the order of 2. This seems however lower than what is usually reported for the evaporation line in terrestrial surface water, but could be akin to it.

*L178-180 From personal experience, I doubt that in-vial evaporation causes drift in the instrument. In such a case, the effect would also be visible in the same manner also by IRMS analysis that uses for example 12mL Exetainers. This is definitely not the case. Thus, possible but very unlikely.*
**Au**: Indeed, we also believe that this is unlikely, as mentioned. However, we use much smaller vials (~2 ml) in CRDS than the Exetainers for IRMS, so such an effect would be much

larger in CRDS than IRMS analyses. Furthermore, we systematically find condensed droplets on the inside of the vial cap when removing the vials after analysis. This is associated with fractionation; however, this likely only induces a small drift. We have made it clear in the revised version of the paper (l. 213)

*L268-290 According to Table 1 the most negative internal standard has a $d_{18}O$ value of -6.61permil, the highest +1.22permil. With respect to the VSMOW-SLAP*
*scale, this is a rather small coverage of the range ($Dd_{VSMOW-SLAP}$ = 55 permil).*
*Yes, seawater has a rather narrow range of isotope values compared to*
*continental freshwaters or precipitation, however, using reference materials so*
*close to each other will not improve the overall precision of the measurement.*
*For scale normalization you calculate a regression line through 2 or 3 of your*
*standards and if isotope values are rather close any scatter in the 2 to 8 values*
*used for the average calculation will already shift your slope and intercept of*
*the line. This effect should be less problematic for reference materials that*
*show a wider separation in isotope values. Also, the other laboratories might*
*have used a different range of reference materials. Seawater isotope analysis is*
*somewhat different to groundwater or precipitation, which covers a far wider*
*range of isotope values (and the respective standards).*
**Au**: We are aware that it is unusual to use such a small range. We also agree that the slope estimate will be less accurate, on the other hand this does not result in a larger error. Our choice was motivated by the possibility of non-linearities in the relationship between the CRDS output and the isotopic delta value. With that in mind and as we apply a linear fit, it is wiser not too use a too large range to minimize the potential bias. On the other hand, we have not detected non-linearities when running a larger set of standards and substandards over a wider range such as VSMOW-SLAP. We were aware of possible differences with laboratories that would be more tuned to analysis of groundwater and precipitation, which is why we also did numerous intercomparisons of CISE-LOCEAN internal standards with other laboratories (that are presented in the paper or in earlier papers).

*L312 The "salt isotope effect" might need a deeper discussion here. First, this effect*
*only influences IRMS measurements done by equilibration ($CO_2$ and $H_2$) but*
*does not apply to LAS (Picarro, Los Gatos). So, theoretically laser*
*measurements should not be corrected in any manner. This not so clear from*
*the text. You apply a correction approach by Benetti et al (2017) who claim that also LAS*
*$d_{18}O$ data (at least Picarro) should be corrected by 0.09‰ due to*
*incomplete evaporation in the vaporizer.*
*Second, the effect of seawater salinity to the chemical equilibrium (activity vs*
*concentration scale) has been tested and discussed in quite a few publications.*
*However, studies were never 100% conclusive. This correction is reported to*
*be –0.15‰ for $\delta_{18}O$ (Lecuyer et al, 2009) and -2‰ for dD (Martineau et al.*
*2012) between the fresh water reference materials and the saline samples.*
*However, other studies did not observe such an effect for seawater salt*
*concentrations (Horita et al., 1993a,b; Bourg et al., 2001). Consequently, it is*
*mainly unclear for most datasets if or if not a correction has been applied to the*
*final measured value. Is this stated for the LOCEAN data in the database what*
*correction have been applied to older IRMS data? This is also not really clear*

*from the text.*

*Bourg, C., Stievenard, M. and Jouzel, J. (2000), Hydrogen and oxygen isotopic composition of aqueous salt solutions by gas–water equilibration method. - Chemical Geology, 173, 331-337.*

*Horita J., Cole D. R., and Wesolowski D. J. (1993a) The activity-composition relationship of oxygen and hydrogen isotopes in aqueous salt solutions: I. I. Vapor-liquid water equilibration of single salt solutions from 50 to 100°. - Geochimica Cosmochimica Acta, 57(19), 2797-2817*

*Horita J., Cole D. R., and Wesolowski D. J. (1993b) The activity-composition relationship of oxygen and hydrogen isotopes in aqueous salt solutions: II. Vapor-liquid water equilibration of mixed salt solutions from 50 to 100°C and geochemical implications. - Geochimica Cosmochimica Acta 57(19), 4703-4711*

*Lécuyer, C., Gardien, V., Rigaudier, T., Fourel, F., Martineau, F. and Cros, A. (2009): Oxygen isotope fractionation and equilibration kinetics between CO2 and H2O as a function of salinity of aqueous solutions. - Chemical Geology, 264, 122-126.*

*Martineau et al. (2012). D/H equilibrium fractionation between H2O and H2 as a function of the salinity of aqueous solutions. Chemical Geology 291 (2012) 236–240*

**Au**: this is a very interesting point, which would require further investigation. Indeed, there is often some uncertainty on what is reported. Here, we have applied the change of scale by Benetti et al (2017a) for CRDS measurements when distilling the water samples or with salt liners, which indeed may result from incomplete evaporation. This was checked when using new salt liners, after the publication of Benetti et la. (2017a). We have also applied an adjustment for older IRMS data of LOCEAN, supported by Marion Benetti's work, and the small set of comparisons that we had done when starting the CRDS measurements at LOCEAN in 2011-2012. There is certainly some uncertainty on the conclusions one can draw from such limited comparisons, and we agree that the effect of the change of scale from activity to concentration for IRMS data (when equilibrating with CO2 or H2) is still an uncertain one (this is now explicitly mentioned in the paper, but we preferred not to discuss further in this paper the scale change). Furthermore, notice that there seems to be an overall consistency between the different adjusted estimates from CRDS and IRMS data in table 2. At the least we could not identify differences which would be related to inconsistencies in these adjustments.

*L340 Has the correction been applied also to the samples with suspected larger evaporation?*

**Au**: For larger suspected evaporation, we have flagged the data as probably bad, and the data have not been adjusted.

*L347-362 If the samples were identical, both measurement techniques (LAS and IRMS) should result in the same values as samples has undergone evaporation and were split into aliquots later? I cannot directly see how this then helps to test the evaporation correction approach based on the relation between S, d-excess and d-values. Or does this paragraph refer to the salt isotope effect above?*

**Au**: during the cruise, the pairs of samples were collected from the same Niskin bottle but were not identical (not the same types of bottles, storage conditions, and analysis time). Unfortunately, some of the LOCEAN samples of that cruise (which cap was not secured with parafilm) had evaporated. It seems that the storage method (crimped top vials) adopted for

the samples analysed at Geozentrum Erlangen was more secure, and those do not appear to have evaporated. This is clearly stated in the revised version of the paper.

*Technical comments*
*L312 found*
**Au**: Done

*L337 space character between delta and D*
**Au**: Done

**Reviewer 2**

*Summary and overall opinion:*

*This study compiles and discusses the uncertainty assessment of seawater del-18O and del-2H. The data set will be very useful for many applications such as constraining ocean circulation. The main issue is with writing and figure captioning and labelling, some of which I have listed below. Overall, I recommend this manuscript for publication after authors edit it thoroughly for grammar and phrasing.*

**Au**: thank you for your overall appreciation and your comments. We have thoroughly reread the paper and have made all the editorial changes that are listed below, including on the figures.

*Minor comments:*

**Abstract:**

*Line 8: It'd be better to change "0.03 / 0.15‰ for δ18O and δD." to "0.03 and 0.15‰ for δ18O and δD, respectively." You can apply this comment to any setup like this in the entire manuscript.*

**Au**: Good idea. We have applied it throughout of the paper.

We have also applied all the changes suggested below to the paper

**Introduction:**

*Line 33: change "designed as" to "termed as".*

*Line 53: you can use one "in" and name all the basins.*

*Line 58: do you mean "some samples were also run"?*

*Line 72: add the "to which" after "extent".*

**Uncertainties:**

*Line 132-133: change "before been analyzed" to "before analysis".*

*Line 139: change "was" to "were".*

*Line 150: change "this way … " to "this technique is applied to minimize the inter-sample contamination …".*

**Au**: note that we have changed the original sentence to "This technique is applied to minimize contamination from the previous sample, even though such memory effect should be small, in particular for $\delta^{18}O$" in order to make sure that the explanation is clear and complete.

*Line 182: change "sea water" to "seawater".*

*Line 219: change "As commented above" to "As mentioned above".*

*Figure 1: Please add panel labels (a, b, c, and d) and axis labels to the figure.*

*Figure 3: add labels to each panel. Also you do not need to say "six maps". Readers can see the number of maps. Just label them and caption each label separately. Also add axis label to the color bars.*

*Figure 4: add label to the color bar.*

*Figure 5: same comments as above.*

**Citation**: https://doi.org/10.5194/essd-2022-34-RC2

**Au:** all suggested changes were incorporated in the revised manuscript.

Example of changed labels on figure 3, and alternative proposition for figure 3.

[Figure]

An alternative version for figure 3 would be to just have one panel (the whole world) with a slightly different colour scale, which we attach for your appreciation (of course, we would add a title and the variable name $\delta^{18}O$ on the color bar). Our current preference is to stick with the first option with six panels.